# Microstructure Evolution and Mechanical Behavior of Mo–Si–N Films

**Yu-Cheng Liu, Bing-Hao Liang, Chi-Ruei Huang and Fan-Bean Wu \***

Department of Materials Science and Engineering, National United University, Miaoli 36063, Taiwan;
com35518617@gmail.com (Y.-C.L.); binghaoliang727@gmail.com (B.-H.L.); zx8110256zx@gmail.com (C.-R.H.)
* Correspondence: fbwu@nuu.edu.tw; Tel.: +886-37-382232

**Abstract:** The molybdenum silicon nitride (Mo–Si–N) films were deposited by a radio frequency (RF) magnetron reactive dual-gun co-sputtering technique with process control on input power and gas ratio. Composition variation, microstructure evolution, and related mechanical and tribological behavior of the Mo–Si–N coatings were investigated. The $N_2/(Ar + N_2)$ flow ratios were controlled at 10/20 and 5/20 levels with the tuning of input power on the Si target at 0, 100, and 150 W. As the silicon contents increased from 0 to 33.7 at.%, the film microstructure evolved from a crystalline structure with $Mo_2N$ and $MoN$ phases to an amorphous feature with the $Si_3N_4$ phase. The analysis of selected area electron diffraction patterns in TEM also indicated an amorphous feature of the Mo–Si–N films when Si content reached 20 at.% and beyond. The hardness and Young's modulus changed from 16.5 to 26.9 and 208 to 273 GPa according to their microstructure features. The highest hardness and modulus were attributed to nanocrystalline $Mo_2N$ and $MoN$ with Si solid-solution. The crystalline Mo–Si–N films showed a smooth tribological track and less wear failure was found. In contrast, the wear track with severe failures were observed for Mo–N and amorphous Mo–Si–N coatings due to their lower hardness. The ratios of $H/E$ and $H^3/E^2$ were intensively discussed and correlated to the wear behavior of the Mo–Si–N coatings.

**Keywords:** Mo–Si–N; magnetron sputtering; microstructure; hardness; wear

---

## 1. Introduction

Surface technologies have been widely applied in enhancing the required performances for specific properties, such as adhesion, hardness, and tribological and corrosion resistance characteristics, for many industrial applications. Hard coatings especially are adopted in versatile industries as protective coatings because of their specific behavior lifetime, and excellent protection features. In late decades, transition metal nitrides (TMN) films have been frequently applied as protective coatings to promote component properties and to prolong the service life in versatile aspects, such as drilling, cutting, molding, tribological resistance, and functional surfaces [1–4]. Material systems, like TiN [5], CrN [6], TaN [7], and MoN [8], and their dual or even multinary nitride layers [9–13], are intensively studied to meet modern demands. Amongst the frequently used nitrides, the Mo–N film draws attention because of its sufficient hardness and Young's modulus, and relatively lower friction coefficient. The ability to form a self-lubricating layer of molybdenum oxide during wear tests has made Mo–N a promising coating system for tribological application [14–19]. Moreover, the incorporation of Si in TMN films generally leads to the formation of nanocomposite TM–Si–N films, in which TMN nanocrystalline grains are solid-solution strengthened or are surrounded by an amorphous $Si_3N_4$ phase [19–28], and better characteristics are realized subsequently. For example, Lin et al. produced Cr–Si–N nanocomposite films by a closed field unbalanced magnetron sputtering system with Cr and Si dual targets [20]. The Cr–Si–N film microstructure evolves from CrN solid-solution with Si,

nanocrystalline Cr(Si)N, and to a dual phase structure composed of CrN crystallite embedded in amorphous $Si_3N_4$, when Si in Cr–Si–N film increases from 0 to 10.2 at.%. The maximum hardness of 38 GPa, optimized $H/E$ and $H^3/E^2$ ratios of 0.096 and 0.31 GPa, respectively, were obtained for the coating with 6.7 at.% Si. Liu et al. fabricated the W–N and W–Si–N films using direct current magnetron cosputtering and investigated the microstructure, mechanical properties, and oxidation behavior of the films. The crystalline W–N coating transformed into an amorphous W–Si–N coating due to the formation of $Si_3N_4$, $W_2N$, and W constituents [23]. Encouraging results concerning various TMN films with Si insertion have been reported in the last decade for enhanced protective properties [29–33]. It is thus persuaded that the incorporation of Si in TMN film works as an effective approach to control the microstructure feature of TM–Si–N films and their characteristics as well.

With the efforts in development of TMN films in consideration, the Mo–Si–N coating is recognized as a good example of such a composite microstructure and enhanced properties [9]. The Si incorporated Mo–N system not only shows superior hardness but also exhibits a decreased friction coefficient, cracking propagation resistance, and enhanced tribological behavior [1–4,19,28,34,35]. In the previous work [9], Lin et al. successfully fabricated the Mo–N and Mo–Si–N multilayers and investigated the mechanical behavior of various Mo–Si–N layers. The microstructure and related properties, as a function of fabrication gas flow and input power, were investigated. The parameter of the $Ar/N_2$ ratio was fixed at 12/8, while the input powers on Mo and Si targets were fixed at 135 W and controlled from 0 to 150 W, respectively. However, the research goals focused on silicon codeposition of 10 at.% and less. The evolution of microstructure and mechanical properties of the TMN film as a function of Si addition up to 30 at.% are not fully understood. Therefore, it is one of the key points of this study to explore the optimized condition of Mo-Si-N sputtering parameters for high Si amount incorporation. Secondly, the effects of phase and microstructure features on mechanical properties of the cosputtering Mo–Si–N coatings are of high interest. In this work, the ternary Mo–Si–N coatings were carried out through a radio frequency reactive magnetron sputtering technique. The process controlling parameters of input power on the Si target and gas flow ratio of $N_2/(Ar + N_2)$ were focused. The microstructure evolution as a function of input power and gas flow ratios was explored and elucidated. The composition and microstructure of the Mo–Si–N films were analyzed by field emission electron probe microanalysis (FE-FMA), field emission scanning electron microscopy (FE-SEM), X-ray diffraction (XRD), transmission electron microscope (TEM), and electron spectroscopy for chemical analysis (ESCA) technologies. Hardness, Young's modulus, and tribological behavior were evaluated and discussed with respect to microstructure features.

## 2. Materials and Methods

The Mo–Si–N films were prepared by a dual gun radio frequency magnetron reactive cosputtering system (Jusun Tech, New Taipei City, Taiwan). The Si and Mo targets with a 99.995% purity were employed as the sputtering sources. The AISI 420 steel and Si wafer were chosen as substrates for film deposition. The Si wafer is atomistically flat and easily coated for microstructural investigation. The Mo–Si–N films were deposited onto AISI 420 steel substrate for mechanical property evaluations. The substrates were cleaned and loaded in the sputtering system, followed by evacuation to a vacuum down to $1.1 \times 10^{-3}$ Pa. Argon and nitrogen were then introduced into the chamber as plasma and reactive source gases, respectively. The pure Mo interlayer was deposited at a 100 W input power for 10 min onto substrates before sputtering of the nitride films. For Mo–N and Mo–Si–N deposition, the $N_2/(Ar + N_2)$ gas ratio was controlled at 10/20 and 5/20 sccm/sccm with a working pressure of $6.7 \times 10^{-1}$ Pa. With a fixed Mo input power of 135 W, the input power for Si was controlled at 0, 100, and 150 W. The sputtering time was set from 1.5 to 2 h to obtain a thickness of approximately 1 µm for all coatings. The controlling parameters and respective sample nomenclature are designated in Table 1.

**Table 1.** Sample designation, controlling parameters, composition, and deposition rate of the Mo–N and Mo–Si–N films.

| Sample Designation | Gas Inlet $N_2/(Ar + N_2)$ (sccm/sccm) | Power Input (W) | | Composition (at.%) | | | Deposition Rate (nm/s) |
|---|---|---|---|---|---|---|---|
| | | Mo | Si | Mo | Si | N | - |
| A1 | | | 0 | 66.4 ± 0.8 | 0 | 33.6 ± 0.8 | 0.133 |
| A2 | 10/20 | 135 | 100 | 41.9 ± 1.0 | 13.2 ± 0.4 | 44.9 ± 1.4 | 0.177 |
| A3 | | | 150 | 34.9 ± 0.5 | 19.8 ± 0.2 | 45.3 ± 0.7 | 0.213 |
| B1 | | | 0 | 75.8 ± 0.6 | 0 | 24.2 ± 0.6 | 0.134 |
| B2 | 5/20 | 135 | 100 | 46.5 ± 0.6 | 22.6 ± 0.5 | 30.9 ± 1.1 | 0.146 |
| B3 | | | 150 | 41.1 ± 0.4 | 33.7 ± 0.3 | 25.2 ± 0.6 | 0.268 |

The field emission scanning electron microscopy (FE-SEM, JSM-6700F, JEOL, Tokyo, Japan) was applied to investigate the microstructure and surface morphology of the films. The coating thickness was also measured through the SEM cross-sectional images. A field emission electron probe microanalyzer (FE-EPMA, JXA-8500F, JOEL, Tokyo, Japan) was employed to check the chemical composition of different samples. The transmission electron microscope (TEM, JEM-2100, JEOL, Tokyo, Japan) with selected area electron diffraction technique was utilized for the detailed microstructure and phases of the Mo–Si–N coatings. An X-ray diffractometer (XRD, Ultima IV, Rigaku, Japan) was utilized to identify the film phase. The binding energy of Si was scanned and analyzed to confirm the Si–N and Si-Si bondings in Mo–Si–N films through electron spectroscopy for chemical analysis (ESCA, PHI 5000 VersaProbe II, ULVAC-PHI, Tokyo, Japan). Mechanical behavior, including hardness, Young's modulus, and tribological behavior, of the coatings were investigated and discussed. Nano-indentation (TriboIndenter, TI 900, Hysitron, Eden Prairie, MN, USA) was performed to analyze coating hardness and Young's modulus. The hardness and Young's modulus were deduced from the indentation-displacement curves with the Oliver–Pharr method [36]. A Berkovich indenter with a normal maximum load of 5 mN and 10 s dwelling was adopted. At least five indents were made for each sample for statistical calculation. The depth of the indentation penetration was kept around 100 to 120 nm for a reliable hardness and Young's modulus calculation according to the 1/10 rule [36]. A wear tester (A20-339, Jiin-Liang, Taipei, Taiwan), with a linear back-and-forth sliding mode and against $Al_2O_3$ balls with a radius of 3 mm as counterpart, was employed to evaluate the tribological behavior of Mo–Si–N coatings. The loading and sliding distance were set at 3.2 N and 100 m, respectively.

## 3. Results and Discussion

### 3.1. Chemical Composition

The chemical composition of Mo–N and Mo–Si–N films fabricated under various parameters is shown in Table 1. Samples designed with A and B followed by a tracking number of 1, 2, and 3 represent the deposition process under $N_2/(Ar + N_2)$ ratios of 10/20 and 5/20 in sccm/sccm, and input powers on Si target of 0, 100, and 150 W, respectively. The binary Mo–N A1 layer showed an N content around 33.6 at.% and a composition stoichiometry close to $Mo_2N$ was realized when the $N_2/(Ar + N_2)$ ratio was set as 10/20. The decrease in $N_2/(Ar + N_2)$ ratio to 5/20 lowered N contents down to 24.2 at.%, as could be seen for B1 film. This was due to lower $N_2$ flow for the reaction and higher Ar gas and its ion flux to generate larger amounts of secondary electrons and resultant Mo atoms and ions during deposition. The B1 film is off stoichiometry of $Mo_2N$ and excess of Mo was expected. For Mo–Si–N layers, the silicon contents were raised with input power. It was noted that as the input power on the Si target was promoted to 100 W, for example for the A2 sample, the film possessed around 13 at.%. A further increase in Si power to 150 W induced a higher Si content of 19.8 at.%. On the other hand, the lowest $N_2/(Ar + N_2)$ ratio of 5/20 in B series deposition led to insufficient N contents of less than 30.9 at.%. Thus, high Si contents of 22.6 and 33.7 at.% were obtained for B2 and B3 deposits, respectively.

In addition, the deposition rate had an increasing trend in terms of Si input power for both A and B series. Additionally, a higher Si input power provoked the resultant higher deposition rate over 0.2 nm/s and a higher Si content in the Mo–Si–N films for A3 and B3 coatings.

### 3.2. Phase Identification

The X-ray diffraction technique was applied to identify the phase and crystal structure of various Mo–Si–N coatings. The diffraction plots of the films are collected in Figure 1. The strong peaks for A1 at 2-theta of 37° and 42.6° were (111) and (200) facets for the $Mo_2N$ phase, respectively. Minor peaks of $Mo_2N$ (220), MoN (222) and (204), and steel substrates at higher 2-theta angles were observed.

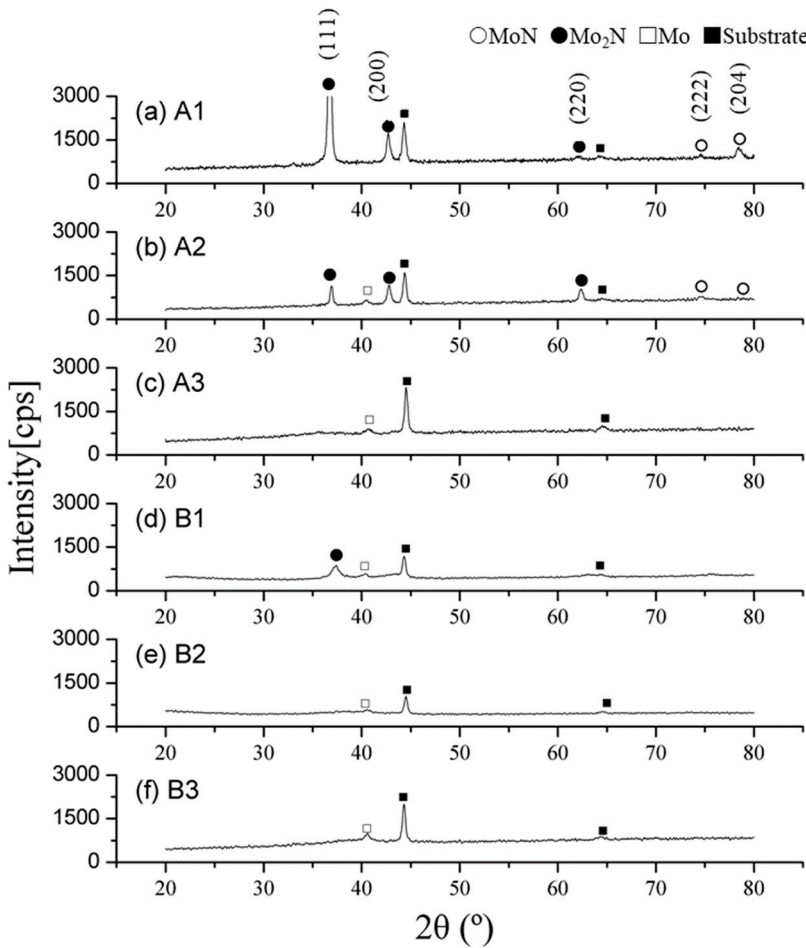

**Figure 1.** X-ray diffraction patterns of the Mo–Si–N coatings fabricated under various gas flow ratios and Si input powers. (**a**) A1; (**b**) A2; (**c**) A3; (**d**) B1; (**e**) B2; (**f**) B3 Samples.

A majority of $Mo_2N$ phases with a limited MoN phase were confirmed. Such a result was in good agreement with the near-stoichiometry of $Mo_2N$, i.e., Mo/N = 66.4/33.6, as depicted in Table 1. With the decrease in the $N_2/(Ar + N_2)$ ratio from 10/20 to 5/20, the intensities of $Mo_2N$ diffraction peaks descended and the reflections for the MoN phase disappeared, as shown in Figure 2d. This implied a weakened $Mo_2N$ phase when larger amounts of Mo and fewer N were deposited, leading to a significant off-stoichiometry of $Mo_2N$ for the B1 sample, as can be seen in Table 1. Moreover, the decrease in peak intensity of the $Mo_2N$ phase while the $N_2/(Ar + N_2)$ ratio decreased also implied a reduction in grain size and crystallinity. For A2 film, $Mo_2N$ and MoN phases remained, however the peak intensities reduced significantly when 13.2 at.% Si was incorporated, as shown in Figure 1b. Further increase in Si content to 19.8 at.% of the A3 coating showed limited peak intensity, implying an amorphous microstructure feature. Likewise, for B2 and B3 films, the Si contents in Mo–Si–N reached 22.6 and

33.7 at.%, respectively. Limited diffraction counts for both $Mo_2N$ and MoN phases and the resultant amorphous feature was expected, as shown in Figure 1e,f.

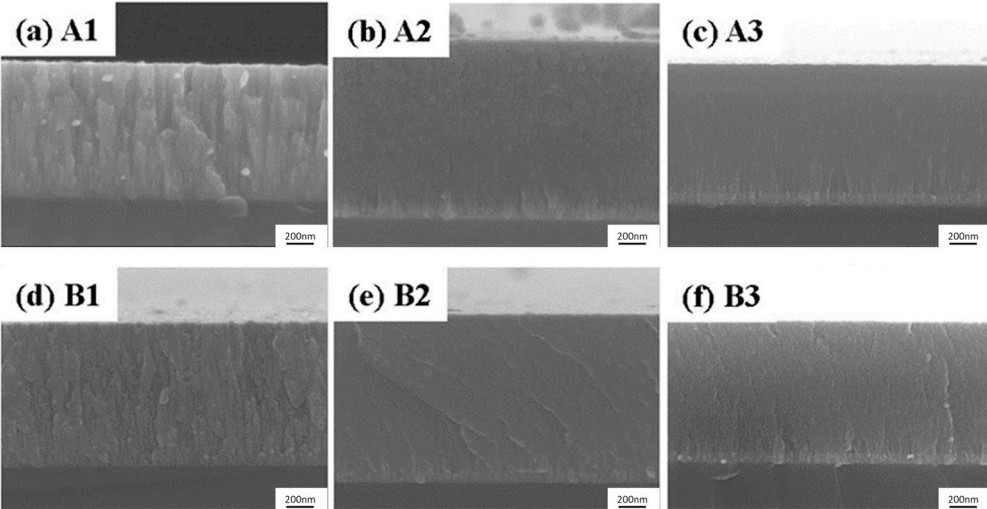

**Figure 2.** Cross-sectional SEI of various Mo–N and Mo–Si–N films. (**a**) A1; (**b**) A2; (**c**) A3; (**d**) B1; (**e**) B2; (**f**) B3 samples.

*3.3. Microstructure*

The microstructure evolution of the Mo–Si–N layers as a function of the deposition parameter can also be observed through SEM cross-sectional morphology in Figure 2. The columnar crystalline feature was found for A1 and B1 MoN binary coatings, as illustrated in Figure 2a,d. The images coincided with the results in X-ray diffraction analysis. When Si contents were intensively added in the Mo–Si–N films, the crystalline structure evolved into the amorphous feature. As shown in Figure 2c–f, no crystalline feature could be recognized in the A3, B2, and B3 samples. This indicated that the increase in Si contents would suppress the $Mo_2N$ and MoN phase formation and growth. Especially, for a lower $N_2/(Ar + N_2)$ ratio of 5/20 and high Si contents of around 20 at.% and above, an amorphous structure formed and dominated in the Mo–Si–N coatings.

In order to have a detailed look into the effects of deposition parameters on microstructure evolution, the cross-sectional TEM image, SAED patterns, and dark field images for various coatings were analyzed. Figure 3 plots the results of A1 and B1 samples. The columnar crystalline structure of the binary Mo–N A1 and B1 films without Si doping is found in Figure 3a,d, respectively. The SAED patterns also confirmed the $Mo_2N$ (111), (200), (220), (311) and MoN (222), (204) phases in the A1 and B1 films, as illustrated in Figure 3b,e, respectively. In Figure 3c, the dark field images of the selected arrowed diffraction were taken to show the crystallography. Columnar growth along film deposition direction was observed for A1. It should be mentioned that under a lower $N_2/(Ar + N_2)$ ratio of 5/20, the B1 sample showed a relatively finer grain feature as compared to that of A1, as shown in Figure 3d. The ring pattern in Figure 3e was also evident for the fine microstructure. This was in good agreement with the results in X-ray diffraction phase analysis.

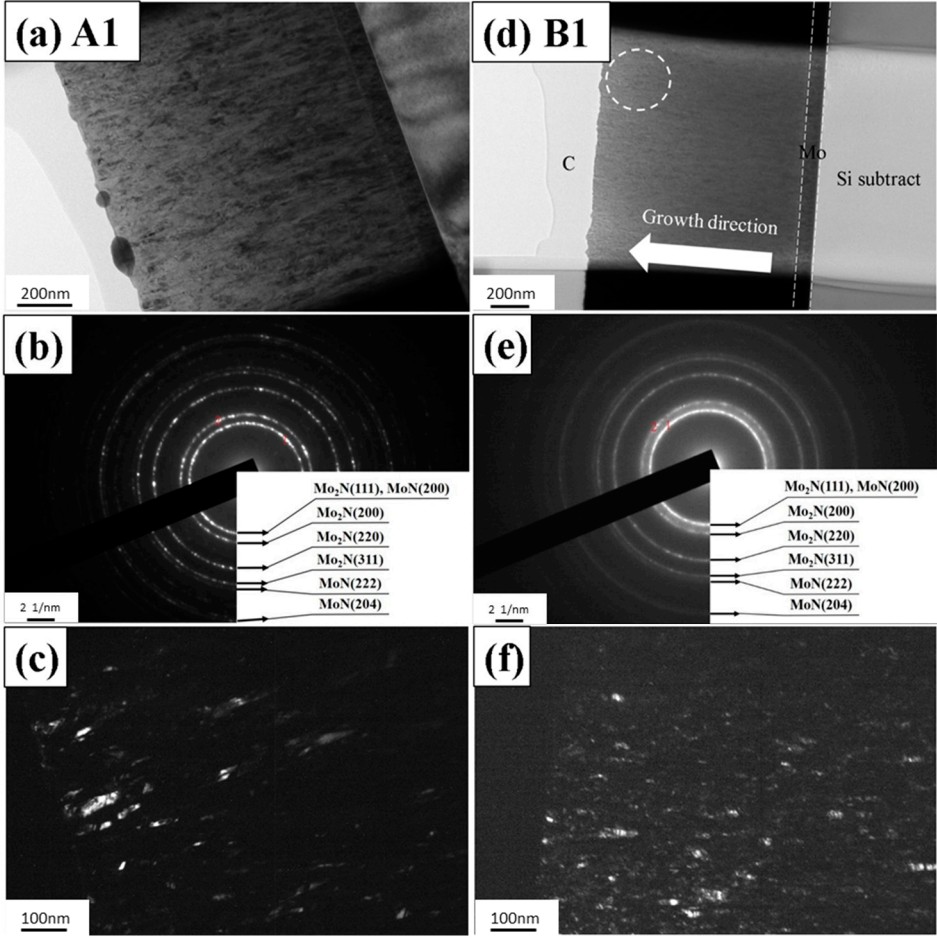

**Figure 3.** The cross-sectional TEM images (**a**,**d**); the SAED patterns (**b**,**e**); and dark field images (**c**,**f**); for A1 and B1 coatings, respectively.

As Si input power was set at 100 W, A2 and B2 samples showed a columnar and an amorphous structure, respectively, as can be seen in Figure 4. The A2 sample possessed a strong columnar structure with $Mo_2N$ (111), (200), (220), (311) and MoN (222), (204) phases, as shown in Figure 4a–c. On the other hand, the B2 layer with a higher Si content of 22.6 at.% exhibited a featureless microstructure and no crystalline phase could be observed in Figure 4d,f. The diffused ring pattern in Figure 4e, which indicated an amorphous structure, also confirmed this point. When Si input power was set to 150 W, i.e., for A3 and B3 samples, the microstructure evolved to an amorphous structure, as depicted in Figure 5. The blurred ring patterns in Figure 5b,e again implied the amorphous microstructure feature of the Mo–Si–N coatings. With a high Si doping, the crystalline feature was postponed, and the amorphous microstructure was evident for A3 and B3 samples, as shown in Figure 5c,f.

### 3.4. Chemical Bondings

The binding energy analysis of Si for various Mo–Si–N films was performed and shown in Figure 6. The peak at 101.5 eV reflected the chemical Si–N bonding of the $Si_3N_4$ phase in Mo–Si–N, while that at 98.6 eV represented the Si–Si bond. With 13.2 at.% Si incorporation, i.e., the A2 sample, the Si–N peak shifted toward the lower energy side and the peak intensity was limited. On the other hand, coatings with high Si contents over 19.8 at.%, i.e., A3 and B3, showed significant signals for the $Si_3N_4$ phase and no peak for Si–Si was found. For the A2 sample, the Si–Si peak at 98.6 eV was proof of Si in a non-nitrided state. The unreacted Si in B2 contributed to the amorphizing of film structure and an intensive diffused ring was thus observed for the B2 sample, as confirmed in TEM analysis

in Figure 4d,f. The major peaks of Si–N for A3 and B3 samples in Figure 6 stood for the $Si_3N_4$ phase formed in the Mo–Si–N coatings when a highest Si input power of 150 W was applied. The amorphous feature of these films with intense Si addition and the amorphous $Si_3N_4$ phase formation were again manifested through ESCA.

### 3.5. Surface Hardness and Elastic Modulus

The hardness and Young's modulus of Mo–N and Mo–Si–N films were investigated and shown in Figure 7. The values of hardness and modulus ranged from 16.5 to 26.9 GPa and from 208 to 273 GPa, respectively, as summarized in Table 2. The hardness of binary Mo–N coatings, i.e., A1 and B1 films, showed hardness of 19.1 and 16.5 GPa, respectively. In the composition and X-ray phase analyses, the A1 sample exhibited a composition close to the $Mo_2N$ stoichiometry and a strong $Mo_2N$ phase. On the other hand, the B1 film possessed the lowest hardness due to less N and excess Mo contents, implying a less hard $Mo_2N$ phase was formed in B2 film. The highest hardness and Young's modulus of 26.9 and 273 GPa, respectively, were obtained for the A2 sample, while average hardness and Young's modulus for the rest were approximately 17–18 GPa and 220–230 GPa, respectively. It should be noted that the 13.2 at.% Si added Mo–Si–N film, i.e., A2, with a columnar crystalline structure presented a higher hardness value of 26.9 GPa. The nanocrystalline feature of the $Mo_2N$ phase with adequate Si doping in Mo–Si–N films were the key factors in strengthening. Moreover, the Si–Si bonding feature found in ESCA for the A2 film also indicated a Si solid-solutioning in $Mo_2N$ and MoN phases. A resultant higher hardness of the A2 sample, as compared to other films, was then expected. When the Si incorporated was beyond 19.8 at.%, the Mo-Si-N films, including A3, B2, and B3, exhibited a significant $Si_3N_4$ phase and resulted in an amorphous structure. Additionally, the lower hardness values for those coatings were obtained due to the soft $Si_3N_4$ phase. For Young's modulus, the A2 sample, which exhibited adequate Si incorporation and effective solid-solutioning strengthening, had the highest modulus. As one could see from X-ray diffraction patterns and SAED patterns of the A2 sample, as shown in Figures 1b and 4b, respectively, the solid-solutioning of Si into A2 Mo–Si–N film was evident. When a higher amount of Si over 19.8 at.% was added, the modulus of the Mo–Si–N films, A3, B2, and B3, dropped drastically to 208–219 GPa due to the soft $Si_3N_4$ phase. In comparison, the binary Mo–N films without Si doping showed medium values of Young's modulus around 241–260 GPa, since the binary Mo–N films remained a major $Mo_2N$ hard phase. In short, the Mo–Si–N film with a controllable amount of Si around 13.2 at.% exhibited a superior hardness due to nanocrystalline $Mo_2N$ and MoN phases with Si solid-solution. Significant descending in hardness and Young's modulus was caused by a soft $Si_3N_4$ phase and the amorphous feature in Mo–Si–N coatings.

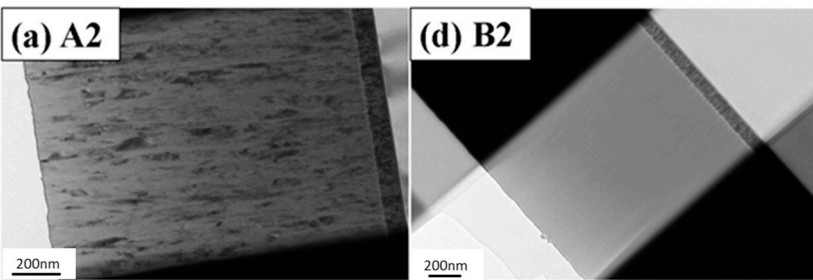

**Figure 4.** *Cont.*

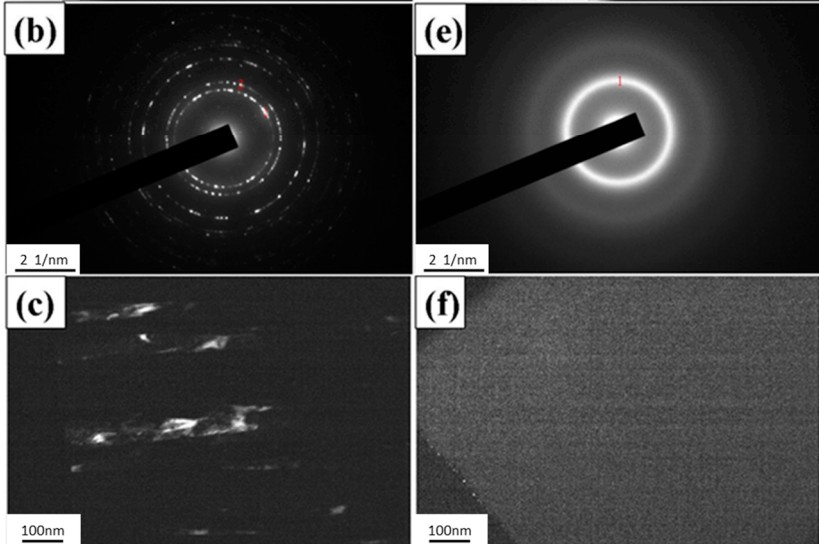

**Figure 4.** The cross-sectional TEM images (**a**,**d**); the SAED patterns (**b**,**e**); and dark field images (**c**,**f**); for A2 and B2 coatings, respectively.

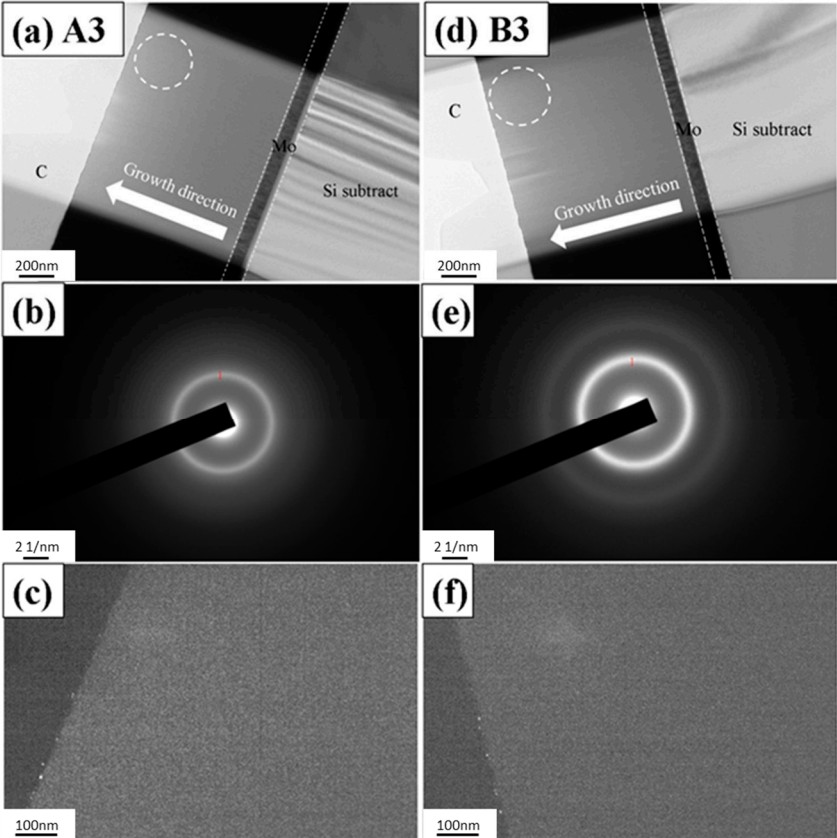

**Figure 5.** The cross-sectional TEM images (**a**,**d**); the SAED patterns (**b**,**e**); and dark field images (**c**,**f**); for A3 and B3 coatings, respectively.

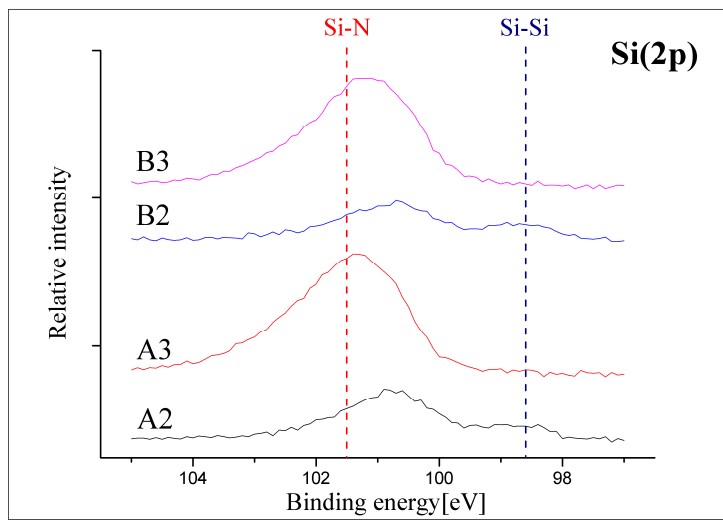

**Figure 6.** The chemical bonding of Si in Mo–Si–N films.

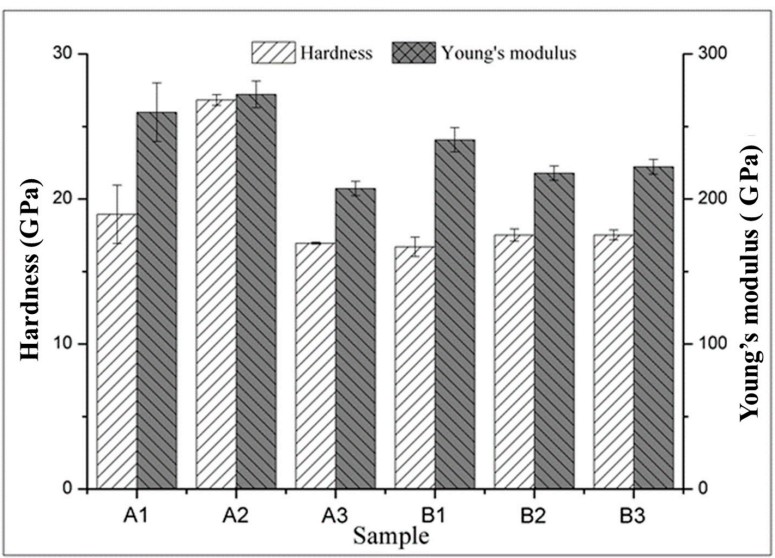

**Figure 7.** Hardness and Young's modulus of various Mo–N and Mo–Si–N films.

**Table 2.** Hardness *H*, Young's modulus *E*, *H/E*, and $H^3/E^2$ ratios of various Mo–Si–N coatings.

| Sample | Hardness, *H* (GPa) | Young's Modulus, *E* (GPa) | *H/E* | $H^3/E^2$ |
|--------|---------------------|----------------------------|-------|-----------|
| A1 | 19.1 ± 2.0 | 260 ± 21 | 0.073 | 0.103 |
| A2 | 26.9 ± 0.4 | 273 ± 9 | 0.099 | 0.261 |
| A3 | 17.1 ± 0.1 | 208 ± 5 | 0.082 | 0.116 |
| B1 | 16.5 ± 0.6 | 241 ± 9 | 0.068 | 0.077 |
| B2 | 17.5 ± 0.5 | 219 ± 5 | 0.080 | 0.112 |
| B3 | 17.5 ± 0.4 | 213 ± 6 | 0.082 | 0.118 |

*3.6. Wear Resistance Behavior*

The wear behavior of the Mo–N and Mo–Si–N sputtering coatings was evaluated through a ball-on-disk linear dry sliding wear test. The optical microscopic images of wear tracks of various films are shown in Figure 8. The wear track widths were also measured for comparison as indicated. For A1 Mo–N binary films, the wear track was smooth with debris stacking along the edges. The width of A1 was around 200 μm. In contrast, B1 Mo–N binary film deposited with higher Mo contents showed a

severely worn surface with deeper galls. It was argued that the relatively lower hardness caused by a less well-crystallized $Mo_2N$ phase of the B1 film was the main reason. It was noticed that under an identical $N_2/(Ar + N_2)$ ratio, the coating deposited under Si input power of 100 W exhibited a narrowest width and a smoother track morphology. For instance, the wear track width of A2 film was smaller as compared to those of A1 and A3. It was believed that the higher hardness and the nanocrystalline structure by adequate Si incorporation in Mo–Si–N films were the strengthening mechanisms for such tribological behavior. Further increases in Si contents triggered the formation of the $Si_3N_4$ phase, which led to a softer amorphous feature in the Mo–Si–N film. Thus, a widened wear track with severe ploughed scars and debris along the track edges was observed in the A3 sample, as shown in Figure 8c. Similar trends of larger wear scar width and severer tribological failure found for B1 and B3 could be seen in the B-series coatings. As shown in Figure 8, the A2 and B2 films exhibited narrower track widths and superior wear resistance than those without Si and with higher Si contents. Besides, as higher Si content was introduced in the Mo–Si–N films due to higher input power of 150 W, the soft $Si_3N_4$ phase formed and led to significant failure for Mo–Si–N coatings under wear. Another indication on the prediction of the wear resistance could be realized from the hardness ($H$) versus Young's modulus ($E$) and combinatorial index ($H/E$ or $H^3/E^2$), as listed in Table 2. The $H/E$ is a plastic index parameter and $H^3/E^2$ works like an indicator of elastic/plastic behavior [36–38]. In Table 2, $H/E$ and $H^3/E^2$ of various Mo–Si–N coatings are calculated and summarized for comparison. It was noticed that the binary Mo–N coatings showed relatively low values of $H/E$ and $H^3/E^2$, 0.068–0.073 and 0.077–0.103, respectively, meaning that the Mo–N films represented less resistance to plastic deformation under external mechanical contact forces. This explained the severer wear behavior of Mo–N films. For A3, B2, and B3 coatings, all the $H/E$ and $H^3/E^2$ values were close to each other and ranged from 0.080 to 0.082 and from 0.112 to 0.118, respectively. The ploughed scars in the widened wear track for these coatings were expected. In contrast, the A2 film with a solid-solution strengthened feature showed a narrower and smoother track morphology, since the hardness to elastic modulus ratios were relatively higher than the rest of the samples, as indicated in Table 2.

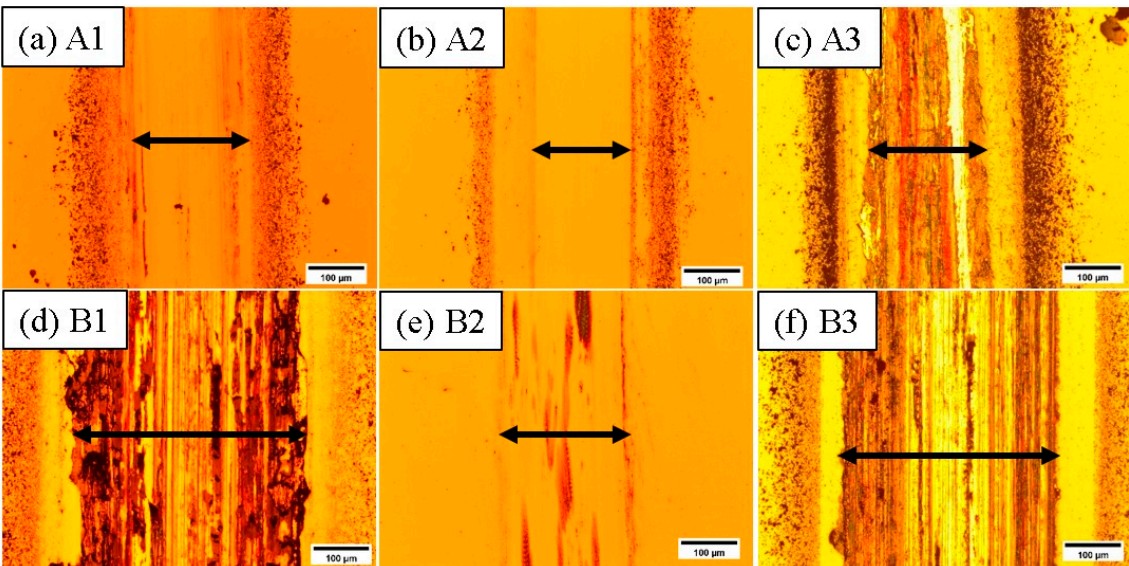

**Figure 8.** The wear tracks of various Mo–N and Mo–Si–N films (**a**) A1, (**b**) A2, (**c**) A3, (**d**) B1, (**e**) B2, and (**f**) B3.

## 4. Summary

The Mo–N and Mo–Si–N films were manufactured by radio frequency magnetron reactive dual-gun co-sputtering with Si input power and a $N_2/(Ar + N_2)$ flow ratio control. The decrease in the $N_2/(Ar + N_2)$ flow ratio resulted in lower N and higher Mo contents, while the promotion in Si input

power provoked higher Si composition and a faster deposition rate. The crystalline films showed diffraction peaks at 37° for $Mo_2N$ (111) with preferred orientation of fcc-Mo(Si)N grains with minor amounts of $Mo_2N$ (200), and (220) and MoN (222), and (204) orientations. The formation of $Mo_2N$ and MoN phases in Mo–Si–N films were suppressed as the Si increased during the sputtering process. A significant $Si_3N_4$ phase formed due to intensive Si incorporation. A high Si content of over 19.8 at.% and an amorphous feature were obtained for the Mo–Si–N coating as Si power was 150 W, regardless of the $N_2/(Ar + N_2)$ ratio. The structure evolution from columnar to amorphous in terms of Si power and $N_2/(Ar + N_2)$ ratio was confirmed. For mechanical properties, the co-sputtering Mo–Si–N films with a $Mo_2N$ phase and strengthened by solid-solutioning of Si incorporation presented maximum hardness and Young's modulus values of 26.9 and 273 GPa, respectively. The incorporation of Si in the range of 13–19.8 at.% into the Mo–N films improved the wear resistance. However, in those Mo–Si–N films with higher Si contents due to higher Si power input, the amorphous $Si_3N_4$ phase dominated and the hardness and the wear resistance decreased significantly. Hence, severer wear damages, including widened wear tracks, deeper ploughed grooves, and debris, occurred accordingly.

**Author Contributions:** Conceptualization, Y.-C.L. and F.-B.W.; methodology, Y.-C.L.; validation, Y.-C.L. and C.-R.H.; investigation, Y.-C.L. and B.-H.L.; data curation, Y.-C.L.; writing—original draft preparation, Y.-C.L.; writing—review & editing, F.-B.W.; supervision, F.-B.W.; funding acquisition, F.-B.W. All authors have read and agreed to the published version of the manuscript.

**Funding:** This research was funded by Ministry of Science and Technology, Taiwan, under grant number MOST-107-2221-E-239-004-MY3.

**Acknowledgments:** The support from the Ministry of Science and Technology, Taiwan, under contracts Nos. MOST-107-2221-E-239-004-MY3 and FE-EPMA Lab, S. Y. Tsai and J. G. Duh, Instrumental Center, National Tsing Hua University are highly appreciated.

**Conflicts of Interest:** The authors declare no conflict of interest.

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
