# Peer review of "Microstructure Evolution and Mechanical Behavior of Mo–Si–N Films"

_coatings, doi:10.3390/coatings10100987_

Round 1
Reviewer 1 Report
The authors present an interesting work on the manufacture and coating of films of the molybdenum silicon nitride system by varying the plasma power and the reactive gas ratio between N2 and Ar.
The authors attempt to correlate a variation in composition and microstructure (mechanical and tribological behavior) through the advancement of the amorphous phase through TEM and Young's hardness and modulus.
However, the work has aspects that must be taken into account in depth before it can be published and to which we call the attention of the authors:
Minor comments:
- It is highly recommended, that the authors mention how the process of measuring the thickness of the samples was?
- Also, what was the selection criteria to select AISI 420 steel and Si wafer as substrates?
- In several sections of the text, "x-ray" appears, it is highly recommended to correct it to X-ray.
- The authors can better explain how they made Young's modulus measurements. What approach did they use in this case (Pharr and Oliver)?
- What values of the modulus of elasticity were found?
- Were the effects of the Ratio of Hardness to Young's Modulus considered? How were they correlated with the variances in deposit pressure?
- Was the influence of the substrate considered in Young's module measurements?
- The authors can explain in depth the sample for the statistical calculation process for the nanoindentation carried out.
Major Comments:
The work is lacking in novelty. The authors have recently published (Lin, Zheng-Xin, et al. "Input power effect on microstructure and mechanical properties of MoSiN multilayer coatings." Surface and Coatings Technology 383 (2020): 125222.) a more in-depth study so that This new adaptation of the results is not very original and has mostly already been discussed in the previous publication.
What is the real contribution of novelty with this work? The authors mention that they have focused on silicon co-deposition greater than 30%. Although it is of interest to study the high incorporation of Si in ternary systems of Mo-Si-N, what is the limit of Si in this system?
In the section "Chemical composition," the authors focus on transcribing the compositional results of the table. However, the stoichiometry obtained in each case is not mentioned. The influence of plasma in a reactive environment (more nitrogen content) and its effect on the excitation of more secondary electrons (less nitrogen content) are not correlated. How is the influence of ion-rich plasma with the tribological properties obtained?
What is the influence of confining the plasma with more Argon?
Table 1:
As expected, in an atmosphere with less available Nitrogen content in the atmosphere, the Nitrogen content in the film decreases. Why is it not only the Ar / N2 ratio?
In the case of Mo, which has always had the same power 135W, the authors report that for the Mo: N relationship it is:
Gas ratio N / (Ar+N) |
Sample |
Power input (Watt.) |
Mo:N |
Si:N |
1/2 |
A1 |
0 |
1,97 |
0 |
A2 |
100 |
0,93 |
0,29 |
|
A3 |
150 |
0,77 |
0,43 |
|
|
|
|
||
5/20
|
B1 |
0 |
3,13 |
0 |
B2 |
100 |
1,50 |
0,73 |
|
B3 |
150 |
1,63 |
1,33 |
Author Response
Dear Reviewer,
Please find the details in the response letter.
Thanks a lot for your time and efforts.
Sincerely,
F.B. Wu

Reviewer 2 Report
The present manuscript deals with a very interesting topic, i.e. the microstructure evolution and mechanical behavior of Mo-Si-N Films. The manuscript may be published upon minor revisions:
- please add some more recent references in the Introduction section;
- in Figure 1 (upper spectrum), labels identifying the (hkl) planes are superimposed to the spectrum therefore it should be redrawn, accordingly;
- please redraw Figures 2 and 3 since the reported bars are too small;
- dark-field images in Figures 3c,f, 4c,f and 5c,f are not so clear, please report micrographs with a higher resolution;
- within the whole manuscript, lots of mistakes are present. Therefore, the paper should be entirely revised, in particular focusing on the text formatting, punctuation and English language.
Author Response
Dear Reviewer,
Thanks for the time and efforts.
Please find the attached file for the details.
Sincerely
F.B. Wu

Reviewer 3 Report
Dear Authors,
The submission describes the results of experimental studies on the effect of two main parameters of magnetron sputtering on morphology and mechanical properties of Mo-Si-N film deposited on AISI 430 steel. I am convinced that the article should be published, but it requires introducing important changes for the readers:
- The introduction: section is quite short: I think it can be extended by listing the specific coating applications that are under investigation. A more detailed description of the results (main findings) of publications grouped in brackets, eg [14-19], [19-28] can also be used for this purpose. Such citation of sources does not give the impression of a good literature research.
- lines 43 and 44: I propose to change "and coworkers" to "et al."
3. M and M: according to the journal guidelines, SI units should be used. Therefore, I suggest adding them in parentheses in justified cases, e.g. after Torr.
- Table 1: what are the reasons for the selection of parameter values? Change Watt to W.
- 78: change x-ray to X-ray
- Figure 7: add spaces before units.
- Conclusions - due to the form (no points), I suggest changing the name of this section to "Summary". It will also be an opportunity to analyze the obtained results in depth and compare them with information from the literature. Sources are cited only on the first page of the manuscript, in the following part no attempt was made to confront the results with the state of the art described in the available articles. Only about 30% of the articles cited are from the last 3 years (9/30). I know from experience that it is good to cite the current works from Mdpi publishing house, as it has a positive effect on the readership of the article.
Author Response

(The authors gave the same response as above.)

Round 2
Reviewer 1 Report
In this new version, the authors have incorporated most of the comments proposed in the first round of review. However, certain aspects still need to be clarified:
- The authors have added the paragraph: "For Young's modulus, the A2 sample, which exhibited adequate Si incorporation and effective solid-solutioning strengthening, had the highest modulus. When (a) higher amount of Si over 19.8 at.% Was added, the modulus of the Mo-Si-N films, A3, B2, and B3, dropped drastically to 208-219 GPa due to the soft Si3N4 phase. The binary Mo-N films without Si doping showed medium values of Young's modulus around 241-260 GPa , since the binary Mo-N films remained a major Mo2N hard phase. "
Authors can correlate this high incorporation of Si in the TEM images, SAED patterns, and dark field images. This is particularly important due to the importance of the very hard Mo2N phases that the authors have reported (241-260 GPa).
Finally, the novelty aspect of the work continues to attract attention. Although I thank the authors for the clarifications, I consider it clear that this article is the continuation of previous work.
On the other hand, the authors affirm that:
"From the viewpoint of materials science and engineering, there will be no limit of Si in Mo-Si-N film. Nevertheless, the Si-N would be too soft for protective purpose and a doping limit of 30-40 at.% ( as one can see in this study that over 20 at.% soft Si3N4 forms and the
film degenerates). "
It is precisely the competition for Nitrogen between Si and Mo that limits the incorporation of Si in the system. These amorphous zones' contribution in the increase or not in the microstructural evolution is a fascinating topic.
It is certainly a very interesting area of study, and we look forward to reading more future work from the authors in this regard.
Author Response
Dear Reviewers,
Please find the attached file for details.

Reviewer 3 Report
Dear Authors,
Thank you for considering my remarks and completing the manuscript. In the revised article, I didn't find the answer to my question from the previous review: "what are the reasons for the selection of parameter values?". I suggest supplementing the manuscript with information on why such parameter values were adopted (in line 89). Does it result from previous experiences or maybe from the analysis of the literature?
Please check the text in terms of technical and editorial (spaces, typos). References should be additionally formatted: the vol number should be written in italics.
Author Response

(The authors gave the same response as above.)
